

# Overview of chicken embryo genes related to sex differentiation

Xiaolu Luo, Jiancheng Guo, Jiahang Zhang, Zheng Ma and Hua Li

Guangdong Provincial Key Laboratory of Animal Molecular Design and Precise Breeding, School of Life Science and Engineering, Foshan University, Foshan, Guangdong, China

## ABSTRACT

Sex determination in chickens at an early embryonic stage has been a longstanding challenge in poultry production due to the unique ZZ:ZW sex chromosome system and various influencing factors. This review has summarized the genes related to the sex differentiation of chicken early embryos (mainly *Dmrt1*, *Sox9*, *Amh*, *Cyp19a1*, *Foxl2*, *Tle4z1*, *Jun*, *Hintw*, *Ube2i*, *Spin1z*, *Hmgcs1*, *Foxd1*, *Tox3*, *Ddx4*, *cHemgn* and *Serpinb11* in this article), and has found that these contributions enhance our understanding of the genetic basis of sex determination in chickens, while identifying potential gene targets for future research. This knowledge may inform and guide the development of sex screening technologies for hatching eggs and support advancements in gene-editing approaches for chicken embryos. Moreover, these insights offer hope for enhancing animal welfare and promoting conservation efforts in poultry production.

## INTRODUCTION

Sex determination in chickens is primarily controlled by the genetic makeup of sex chromosomes (*Clinton, 1998*; *Clinton et al., 2012*; *Kuroiwa, 2017*; *Smith, Major & Estermann, 2021*; *Smith & Sinclair, 2004*). There are 39 pairs of chromosomes in chicken somatic cells, consisting of 38 pairs of autosomes and one pair of sex chromosomes. Unlike mammals, avians use a ZZ:ZW heteromorphic sex chromosome system, where females have ZW heterologous sex chromosomes and males have ZZ homologous chromosomes (*Ellegren, 2009*; *Gilgenkrantz, 2004*; *Graves, 2016*; *Nanda et al., 2008*; *Smith & Sinclair, 2004*). The expression of the Z-linked gene *Dmrt1* plays a crucial role in sex determination in chickens, and this sex-determination mechanism is conserved in birds (*Smith et al., 2009*).

The sex of a chicken has a significantly impacts its productivity level; hens are more efficient at egg-laying, while roosters are more efficient at meat production (*Cygan-Szczegielniak & Bogucka, 2021*; *Rondelli, Martinez & Garcia, 2003*; *Yaghobfar, 2001*). However, in breeding enterprises, male chicks that do not meet the production needs will generally end up being culled, leading the ethical and animal welfare issues becoming a growing problem for the poultry industry. This practice, born out of economic considerations, has sparked a growing concern within the industry, highlighting the urgent need for solutions that can reconcile production efficiency with ethical standards. As the

Corresponding authors
Zheng Ma, mz8522@163.com
Hua Li, okhuali@fosu.edu.cn

global demand for poultry products continues to rise, the importance of developing and implementing sex control technology in animal husbandry cannot be overstated. Such technologies not only have the potential to enhance the welfare of male chicks but also to contribute to the sustainability of poultry production.

To address these challenges, significant attention has been directed towards understanding the genetic underpinnings of sex differentiation in chickens. The ability to control or predict the sex of chicken embryos before hatching could revolutionize the industry, reducing the need for culling and improving overall animal welfare. The incubation time of hatching eggs is typically around 21 days, with gonadal differentiation beginning around embryonic day 2 (E2, Hamburger Hamilton Stage, HH6) (*Zhang et al., 2023*). Sexing hatching eggs poses a major challenge in practical production. Early sex determination of embryos and sex control during incubation can reduce production costs, increase efficiency, and prevent the culling of male chicks after hatching due to unsatisfactory production performance (*Busse et al., 2019*; *de Haas, Oliemans & van Gerwen, 2021*; *Gremmen et al., 2018*; *Leenstra et al., 2011*). Several methods have been developed to address this issue, including detecting the expression levels of specific genes in the embryo to determine its sex. This allows for the screening and removal of male embryos, while female embryos continue to be incubated, a practice that has seen commercial application by several companies. For example, In Ovo (https://inovo.nl/), a biotech start-up, has recently achieved a breakthrough in identifying a unique biomarker that distinguishes between male and female embryos at an early stage of development. Based on this discovery, the company has developed a machine called Ella that can quickly extract tiny samples from eggs and screen out female ones at a high speed (https://inovo.nl/ella/). Moreover, eggXYT employs CRISPR to edit chickens to lay sex-detectable eggs (https://www.eggxyt.com/). Male eggs carry a biomarker in their DNA, causing them to emit a bright yellow signal when scanned with an electro-optical device called seXYT. This device can then identify and remove male eggs at the entry point of the hatchery, ensuring that only female eggs are hatched. There is another company called PLANTegg, which combines allantoic fluid from hatching eggs with a reactive medium and performs specific DNA of the sex chromosomes analysis using PCR to identify males and females (https://www.plantegg.de/).

Researching and applying genes that influence sex differentiation can significantly enhance production efficiency and lead to a more balanced male-to-female ratio in the production line. This not only boosts production efficiency but also improves the welfare of male chickens, addressing a critical ethical concern in the poultry industry. Such advancements have the potential to extend their benefits beyond poultry, offering solutions to similar ethical issues in other areas of animal production and paving the way for more humane and sustainable practices across the industry.

The meticulous research into genes affecting sex development in chickens transcends basic scientific inquiry. It represents a crucial step towards a future where the poultry industry can fulfill production demands without compromising animal welfare. By delving into the specifics of these genetic factors, we deepen our understanding of avian biology and contribute to the development of transformative technologies in animal husbandry.

In the following part, we aim to provide a comprehensive and credible reference for further research into the sex differentiation of chicken embryos. Additionally, we seek to advance the study of gene editing technology in chickens by highlighting specific genes that have been demonstrated by previous studies to influence sex development.

This review is intended for researchers who are interested in sex differentiation in chickens and are committed to safeguarding animal welfare, as well as for egg-laying and meat-laying chicken farms and enterprises whose productivity and profitability are affected by sex differences in chickens.

## Survey methodology

A systematic literature review was conducted to evaluate the influence of specific genes on the sex differentiation of chicken embryos. We performed a comprehensive search across multiple scientific databases, including Web of Science (https://www.webofscience.com/wos/), PubMed (https://pubmed.ncbi.nlm.nih.gov/), and ScienceDirect (https://www.sciencedirect.com/), utilizing a targeted keyword strategy that encompassed "chicken," "chicken embryo," "sex genes," and "chicken sex differentiation." Our search parameters were restricted to research articles published from 1990 to the present to ensure the inclusion of contemporary studies.

The selection process involved a meticulous examination of relevant publications to ascertain the contribution of identified genes to the sex differentiation pathway in chicken embryos. We synthesized the findings, presenting a detailed account of the genetic mechanisms implicated in this biological process. Throughout the review, we maintained a rigorous, unbiased approach, ensuring that our synthesis of the literature was exhaustive and devoid of any preconceived notions. This analysis aims to provide a clear and objective overview of the genetic determinants that govern sex differentiation in avian species, with a particular focus on chickens.

## Genes related to sex differentiation in chickens

Chicken sex differentiation is controlled by sex chromosome-linked and autosomal genes. Sex chromosome-linked genes are crucial for sex determination, typically dictating the male or female pathway. Meanwhile, autosomal genes are thought to act as regulators, influencing the activity of sex-determining genes. The sections below categorize these genes based on their chromosomal location and outline their functions in sex differentiation.

### Sex chromosome-linked genes
#### Dmrt1

At the beginning of the 21st century, various hypotheses about the sex-determining mechanism in birds were proposed, including the Z-chromosome dose–effect hypothesis and the W-chromosome dominant-effect hypothesis. *Shan et al. (2000)* reported that *Dmrt1* (doublesex and mab-3-related transcription factor 1) is exclusively expressed in the testis. This finding has been corroborated in subsequent studies (*Yamamoto et al., 2003*; *Zhao et al., 2007*), *Shan et al. (2000)* also predicted that two gene dosages of this gene are necessary for the testis formation in males. Subsequently, an experiment showed that *Dmrt1* expression levels are elevated in sex-reversal chickens (female to male) (*Smith,*

*Katz & Sinclair, 2003*). Its expression was also found to suppress ovarian pathway genes, like *Cyp19a1* and *Foxl2* (*Hirst, Major & Smith, 2018*). Conversely, the downregulation of *Dmrt1* expression levels was detected in sex-reversal embryos (male to female) (*Fang et al., 2013*). Decreased *Dmrt1* expression level in early hatching eggs led to the feminization of male embryos and a significant reduction in the testicular marker *Sox9* at the mRNA level (*Smith et al., 2009*), supporting the Z-chromosome dose–effect hypothesis. Further studies found that *Dmrt1* is expressed before *Sox9*, highlighting the importance of the *Dmrt1* gene for testicular development (*Chue & Smith, 2011*; *Zhao et al., 2010*).

Since then, numerous investigators have verified this hypothesis, and their findings further support critical role of *Dmrt1* in avian sex determination (*Estermann, Major & Smith, 2021*; *Hirst et al., 2017*; *Ioannidis et al., 2021*; *Lambeth et al., 2014b*; *Lee et al., 2021*). An increased expression of *Dmrt1* serves as a significant genetic marker in testicular formation. It has been confirmed that the sex of birds is determined by a dose effect of the Z-linked *Dmrt1* gene, a mechanism that differs from the dose-compensating effect observed in mammals.

### *cHemgn*

A study by *Nakata et al. (2013)* found that *cHemgn* (Z chromosome-linked chicken homolog of hemogen) is not only expressed in hematopoietic tissues but also in the gonads of early male chicken embryos, with mRNA expression levels significantly higher in the male gonads than in the female ones. Furthermore, *cHemgn* expression was observed in Sertoli cells, which could directly or indirectly activate *Sox9* expression. Overexpression of *cHemgn* in female embryos led to the gonads exhibiting bilateral developmental characteristics of male morphology. These findings suggest that *cHemgn* plays a role in gonadal differentiation and can promote testicular development in male embryos.

### *Hintw*

*Hintw* (W-linked histidine triad nucleotide-binding protein) has been identified as a W-linked ovarian determinant gene (*Ayers et al., 2013a*; *Kuroiwa, 2017*; *Smith et al., 2009*). *Nagai et al. (2014)* demonstrated that *Hintw* is a female-specific gene, with all cells in female embryos strongly positive for *Hintw*, whereas male embryos are entirely negative. Similarly, a study by *Sun et al. (2021)* corroborated Nagai's findings, revealing that overexpression of *Hintw* in female chicken embryos led to upregulation of *Foxl2* and *Cpy19a1*, and downregulation of *Sox9* and *Dmrt1*. Furthermore, overexpressing *Hintw* in male chicken embryos inhibited androgen and increased estrogen levels, further establishing the critical role of *Hintw* in promoting female embryo differentiation.

## Autosomal genes
### *Sox9*

*Sox9* (SRY-box transcription factor 9), a central hub gene required to initiate the development of Sertoli cells (*da Silva et al., 1996*), is a hallmark gene of male gonadal development and is significantly upregulated during male gonadal development (*Carre et al., 2011*; *Yamamoto et al., 2003*), as was its expression on duck and quail embryonic gonads (*Takada et al., 2006*). *Sox9* exhibited higher M6A methylation and mRNA expression levels

in male gonads and appears to be closely associated with *Dmrt1* (*Caetano et al., 2014*; *Jiang et al., 2022*; *Li et al., 2022*). Subsequent studies have found that *Dmrt1* induces and activates *Sox9*, which then promotes testicular differentiation when highly expressed in male embryos (*Ioannidis et al., 2021*; *Lambeth et al., 2014b*). Moreover, *Sox9* is activated when *Dmrt1* is ectopically expressed in female gonads, affecting aromatase synthesis and inhibiting female gonadal development. To sum up, *Sox9* was shown to be a gene that affects embryonic male development and is closely related to *Dmrt1*.

### Amh

The anti-Muller hormone (*Amh*) is a member of the Transforming growth factor TGF-$\beta$ superfamily. Studies have shown that *Amh* is involved in sex differentiation and gonadal development (*Bai et al., 2020*; *Oréal et al., 2002*; *Rahaie, Toghyani & Eghbalsaied, 2018*). Studies showed that *Amh* is upregulated in the testis during gonadal development to undergo Mullerian duct degeneration in male embryos (*Ayers et al., 2015a*; *Ayers, Smith & Lambeth, 2013b*; *Piprek et al., 2018*; *Roly et al., 2018*; *Smith, Smith & Sinclair, 1999*; *Yamamoto et al., 2003*). *Lambeth et al. (2016a)* showed that overexpression of *Amh* inhibits the development of the female gonadal cortex, exhibits a partially masculine appearance, and also disrupts testicular development in males. Furthermore, *Amh* is a downstream gene of *Dmrt1* (*Lambeth et al., 2015*) and is more highly expressed in males (*Cutting et al., 2014*; *Tsukahara et al., 2021*). Distinct from that in mammals, *Sox9* does not activate the expression of *Amh* for the reason that *Amh* expression precedes *Sox9* in chickens (*Oreal et al., 1998*). Thus, *Amh* was shown to be a gene that promotes the differentiation of embryos into males.

### Tle4z1

A new gene discovered by *Chen et al. (2022)*, called *Tle4z1* (transducing-like enhancer of split4), can regulate the expression of *Dmrt1*, *Sox9*, and other male marker genes. When the expression of *Tle4z1* interfered, the gonad development of male chicken embryos was inhibited. On the contrary, overexpression of this gene in female chicken embryos results in ovarian atrophy and hinders gonadal development. Simultaneously, the result shows higher expression levels of *Tle4z1* in males than that in females at all ages, with a clear male preference, suggesting that this gene, as a male-biased gene, plays an important role in the male differentiation of chicken embryos.

### Spin1z

According to the study by *Jiang et al. (2021)*, when *Spin1z* (spindlin1-z) was knocked down in chicken embryo, it was detected that the expression of testis differentiation markers *Amh* and *Sox9* was significantly downregulated in chicken male gonads, and the expression of ovarian development regulatory genes *Cpy19a1* and *Foxl2* was significantly upregulated, the opposite trends were observed when *Spin1z* was overexpressed. This finding demonstrates that *Spin1z* plays an important role in the differentiation of embryos into male chicks.

### Hmgcs1

*Shi et al. (2021)* showed that overexpressing and microinjecting *Hmgcs1* (3-hydroxy-3-methylglutaryl coenzyme A synthetase 1) in chicken embryos promoted the expression

of *Sox9* and increased the level of testis hormone in male gonads. They also found that *Hmgcs1* inhibited the expression of *Cyp19a1*, suggesting that *Hmgcs1* can promote the expression of genes related to the differentiation of chicken gonads into males and one of the genes that influence gonadal differentiation in the chick embryo.

### Foxd1

*Foxd1* (forkhead box D1) has a clear male specificity in embryonic gonads and localizes in the Sertoli cells of the chicken testis (*Yu et al., 2019*). In this study, *Foxd1* was knocked down in Chicken Sertoli cells, and the expression of *Amh* and *Sox9* was significantly downregulated, but the expression of *Dmrt1* remains unchanged, which indicated that *Foxd1* was an important upstream gene of *Sox9* and affected the differentiation of chicken embryonic male gonads.

### Tox3

*Tox3* (TOX high mobility group box family member 3), was shown to be expressed at much higher levels in male chicken embryos than in female chicken embryos during gonadal development, particularly in the developing seminiferous cords of the gonadal medulla (*Estermann, Major & Smith, 2023*). Moreover, it was proved that *Tox3* is activated by *Dmrt1* induction, with localized loss of *Tox3* protein expression when *Dmrt1* knockdown in male gonads, while *Dmrt1* misexpression in females causes upregulation of *Tox3* expression. This result suggests that *Tox3* also plays an important role in the differentiation of gonads into males.

### Cyp19a1

*Cyp19a1* (cytochrome P450 family 19 subfamily A member (1)) encodes an aromatase that is responsible for the conversion to estrogen and is a key gene for the feminization of the gonad in chicken embryos (*Ellis et al., 2012*; *Lambeth et al., 2016b*; *Smith, Andrews & Sinclair, 1997*). It was known that this gene is specifically expressed in females from the beginning of gonadal sex differentiation. Its overexpression in male embryos has been shown to inhibit the expression of three key testis genes–*Dmrt1*, *Sox9*, and *Amh*–leading to the early disappearance of the spermatic cord and thus blocking the male gonads' testis-determining pathway. Furthermore, it upregulates the expression levels of *Foxl2* and *Rspo1* in male embryos, *Foxl2* and *Rspo1* are responsible for ovarian development, promoting female gonadal differentiation and interfering with male gonadal differentiation (*Jin et al., 2020*; *Lambeth et al., 2013*; *Lambeth et al., 2016b*; *Lambeth et al., 2014a*). A study showed that injecting aromatase inhibitors before gonadal sex differentiation in the hatching period of breeder eggs produced more male chicks (*Yang et al., 2009*), a finding that has been validated in subsequent experiments (*Abdulateef et al., 2021*; *Fazli et al., 2015*; *Scheider et al., 2018*).

### Foxl2

Multiple studies have found that *Foxl2* (forkhead box L2) is expressed only in female embryos (*Hudson, Smith & Sinclair, 2005*; *Luo et al., 2020*) and plays a crucial role in regulating ovarian development (*Du et al., 2022*; *Govoroun et al., 2004*; *Wang et al., 2017*;
*Zhang et al., 2019*). Knocking out *Foxl2* in chicken embryos can promote the expression of the gene *Sox9*, while its overexpression in male embryos can antagonize *Sox9* and cause testicular dysplasia, thus affecting the sex differentiation of the embryo (*Major et al., 2019*). Although *Foxl2* is involved in the estrogen pathway of chicken embryos, unlike the mechanism of regulation by activating the *Cyp19a1* gene in mammals, *Foxl2* does not have the capability of activating *Cyp19a1* (*Wang & Gong, 2017*). Despite being co-expressed in the gonads of female chicken embryos, *Foxl2* does not affect aromatase expression in embryonic and sexual maturation (*Guo et al., 2022*).

### Jun

Studies have found that *Jun* is associated with the development of the ovaries and follicles, and its expression in female chicken embryos is higher than that of males (*Ayers et al., 2015b*; *Zhang et al., 2021*). Research by conducted *Zhang et al., (2021)* demonstrated that overexpression of *Jun* in embryos induced female-related characteristics, whereas interference with *Jun* expression resulted in male-like characteristics. Furthermore, it has been suggested that female sex differentiation can be facilitated by promoting the expression of *Cyp19a1* and inhibiting the expression of *Smad2*, a gene localized on the Z-chromosome that promotes embryonic male differentiation (*Li et al., 2022*; *Tagirov & Rutkowska, 2014*).

### Ube2i

A recent study found that a newly identified female-biased gene, *Ube2i* (ubiquitin-conjugating enzyme E2I), is essential for female-specific development in chickens (*Jin et al., 2021*). The overexpression of *Ube2i* led to the upregulation of female-specific genes (*Foxl2*, *Cyp19a1*, and *Hintw*) and downregulation of male-specific genes (*Sox9*, *Dmrt1*, and *Wt1*), indicating that *Ube2i* plays a crucial role in sex differentiation in chicken embryos.

### Ddx4

*Ddx4* (DEAD-box helicase 4) is known to be upregulated solely during ovarian development (*Carre et al., 2011*). When *Ddx4* was knocked down in chicken embryos, there was no effect on the expression of male gonad markers, *Dmrt1* and *Sox9*. However, the expression of female gonad marker *Cyp19a1* was significantly downregulated in the female gonads, and the expression of *Foxl2* also showed a slight decrease (*Aduma et al., 2019*). These findings demonstrate that *Ddx4* plays a crucial role in the differentiation and development of the female gonads in chicken embryos.

### Serpinb11

It was discovered that the expression of *Serpinb11* (serpin family B member 11) mRNA was exclusive to the fallopian tubes of female chickens (*Lim et al., 2011*). Furthermore, the researchers found that subcutaneous implantation of diethylstilbestrol, a drug known to influence the growth, development, and differentiation of the chicken oviduct, led to a remarkable 500-fold increase in the expression of *Serpinb11* mRNA in the fallopian tubes. These findings suggest that this female-specific gene is tissue-specific, playing a crucial role in the growth and development of the chicken oviduct and potentially affecting the differentiation of chicken gonads.

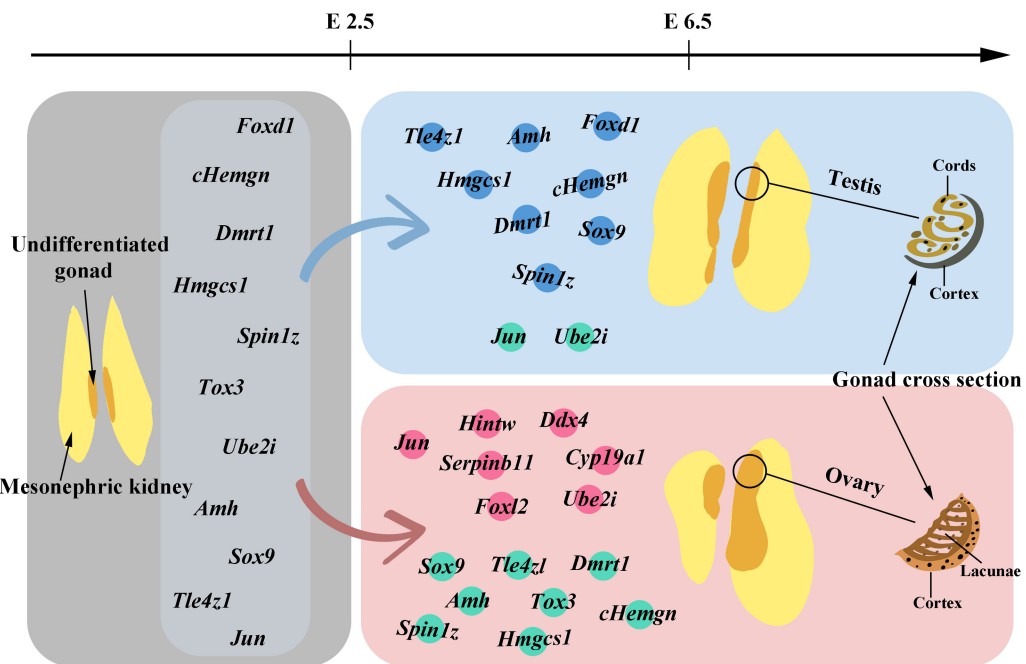

**Figure 1** **Gene expression before and after gonadal development.** The gray area in the figure illustrates the morphology of undifferentiated gonads and also marks the genes detected in these undifferentiated gonads before embryo day (E) 2.5. In gonads differentiating into males, genes with high expression levels from E2.5 to E6.5 are marked with blue dots, while those with lower expression levels are indicated by green dots. For gonads differentiating into females, genes with high expression levels within the same timeframe are labeled with pink dots, and those with lower expression levels are also marked with green dots. Starting from E6.5, the gradual morphological changes in the tissue following gonadal differentiation and development can be observed.

Above, we have discussed the genes that have been identified to influence the sex differentiation of chicken embryos. The following figure (Fig. 1) will provide a brief overview of the gene expression patterns observed before and after gonadal development.

### Candidate genes screened by RNA-Seq

RNA-seq enables the identification of genes exhibiting significant sex differences during the chicken embryonic gonadal differentiation period. In addition to already validated genes such as *Amh* and *Hintw*, numerous unvalidated candidate sex-biased genes have been detected in chicken embryos at E2.5, E4.5, and E6.5 periods, respectively (Table 1). These candidates offer a valuable database and reference for research.

## CONCLUSIONS

The global demand for chicken meat and its by-products is continuously increasing, leading to a significant rise in chicken meat production. However, the practice of sacrificing male chicks from laying hen breeds at birth, due to their substandard production performance, has raised major concerns regarding animal welfare worldwide. Consequently, researching genes that influence sex differentiation in chicken embryos holds immense significance. By

**Table 1  Candidate genes capable of influencing sex differentiation screened by RNA-seq in chicken embryos at day 2.5 (E2.5), E4.5 and E6.5 stages.**

| | Male-biased | Female-biased | References |
|---|---|---|---|
| E2.5 | *NSMF, PGM5, DEF8, FAM234A* | *SMAD7B, HINTW, HNRNPKL, SPIN1W, UBAP2* | *Ichikawa et al. (2022)* |
| E4.5 | *SLC24A2* | *SMAD7B, HINTW, HNRNPKL, UBAP2* | *Ichikawa et al. (2022)* |
| | *NCBP1, LMAN1, TLE4, UBE2R2, MOSPD2, CDC14B, RAD23B, ST3GAL3, SLC30A5, ITGB8, MELK, GNE, ARHGEF3, RIOK2, PPAP2A, UBQLN1, CZH9orf41, SREK1IP1, KIAA1822, SNCAIP, SNX24, TBCA, ADAMTS19, C5orf13, NFIB, NIPSNAP3A, ACAA2, HNRNPK, CENPH, MTAP, SMAD2, UTP15, GRAMD3, CBWD1, ANKRA2, KCMF1, CENPK, DNAJC21, RNUXA, CCDC5, NDUFAF2, ZNF608, SRFBP1, RNF20, TMEM157, CCDC112, ZFR, MAK10, PIK3AP1, TMEM175, AP3B1, PAK1, NNT, ALDH7A1, SMC5, B4GALT1, CZH9orf103, EMB, ZFAND5, CZH9orf3, BXDC2, TMOD1, GNAQ, GNA11, CCNH, PPWD1, PIAS2, AMACR, ZCCHC6, DTWD2, C5orf37, GFM2, MRPL50, C7, ZNF367, MRPS30, HSD17B4, DIMT1L, TARS, RASA1, TTC33, PLAA, WDR70, HDHD2, C18orf10, ZCCHC9, SRP19, HISPPD1, PJA2, RMI1, FANCC, SDCCAG10, WDR32, APCDD1L, NUP155, GOLPH3, SHANK2, ANKRD32, NUDT2, CFC1B, CZH5orf44, YTHDC2, PELO, TOX3, DMRT1, CHD1, CCDC100, SSBP2, TSTD2, ARSK, NIPBL, WDR36, C5orf21, KIAA1797, SPTLC1, FXN, CHSY3, KIF24, PTGER4, NUDT12, LINGO2, RASA1, TUSC1, SPINW, SPINZ, SLC16A7, IKBKAP, HAUS6, KIAA0372, CZH18orf25, CETN3, KIAA0280, INHA, PIP5K1B, FBXO4, FANCG, PPM1H, DIRAS2, PLIN2, PIGG, GRIA2, STARD4, C9orf125, CZH9orf82, CZH5orf42, RNF165, HEMGN, COL14A1, ERCC8, FRMD3, TMEM90A, CPA6, GRM8, VGLL2, SIM1, MAMDC2, AMH, CYBRD1, SERINC5, SLC18A3, VIP, GABRA4, RGSL1, VSTM2A* | *HINTW, RPL17L, RPL17, HNRNPK, ATP5A1W, UBAP2, SMAD2, FAM26A, FOXL2, ZNF532, SOX2, IL28B, MATN3, GRPR, GABRA5, GTSF1, CKMT1A, C1QC, KCND3, TUBA3E, TAGLN3, ADAD1, STK31, VCAM1, AKR1B10, CD9, LUM, HSPA2* | *Ayers et al. (2015b)* |
| E6.5 | *NAV3, FST, MAMDC2, SYTL5, VSX2, AMH, NEU2, MAB21L2, GREM2* | *PRNP, SLC16A4, URAH, SELENBP1, MEOX1, CREB3L3, CYP1A2, CYP1A1, ADRA1B, SLC22A7, ACSM3, CHAT, F13B, ANKS4B, SLC2A5, MAT1A, VAV2, SLC34A1, LGALS2, CDHR2, A1CF, DAB2, UGT1A1, SIGIRR, HNF4A, PLA2G12B, CYP3A4, ASPA, SLC47A2, SLC5A10, BCO1, PNAT3, PNAT10, GJB1, VILL, SUSD2, SLC5A11, SLC13A5, HOGA1, XYLB, DPEP1, SLC25A48, SLC16A12, SLC22A18, FAH, UPB1, SLC5A1, TTC36, MELTF, ENTPD1, APOA1, MASP1, SLC51A, KMO, PCK1, BCO2, CRYAB, AS3MT, XDH, CUBN, SLC13A1, PPP1R1C, GIPC2, FGB, FGA, PLEK2, VEPH1, CLRN3, HAAO, SLC3A1, SLC1A1, OTOGL, AADAC, HPGDS, VDHAP, CYP4B7, PDZK1IP1, SLC5A9, DIO1, AGR2, CTH, UNC93A, AMN, NR1H4, GC, SLC5A8, SLC15A2, EMP1, ACMSD, BAIAP2L2, IYD, CYTH4, FBP2, MARVELD3, SLC6A13, SBK2, DDC, SLC15A5, IAPP, GYS2, PRMT8, CYP19A1, CYB5A, ALDH8A1, VNN1, CSTA, HAO2, MYOM1, GCNT4, HGD, PLA1A, TMEM174, SLC22A16, TTR, PSAT1, TMPRSS7, PDZK1, ALDOB, SH3TC1, HGFAC, FABP1, KCNJ15, TMPRSS2, TINAG, GSTA4, GSTA3, APOB, ACE2, ASB9, CYP2AC1, CLIC5, RGN, RP2, SULT1C3, F7, SLC10A2, GJB6, SLCO2B1, HAVCR1, CLDN10, SLC6A19, AvBD9, KCNK16, CAPN13, ADH6, AGXT, SLC2A11L1, FAM173A, PAFAH2, HOXD4, GATM, CR1L, SUSD3, SLC26A6, PIPOX, PKIB, SLC26A1, ANPEP, LCAT, TM6SF2, SMAD7B, BPHL, PGLYRP2, KRT18, G6PC, AMBP, RASSF2, ADH1C, CYBRD1, APOC3, ENPEP, MYO7A, TMEM98, MC5R, APOH, IFI30, FER1L6, C1orf115, CDH17, HINTW, PLTP, TMEM230, ASTL, SLC23A1, STAC, CYP4F11, HNRNPKL, DGAT2, PTN, FMO3, BAAT, SPIN1W, UBAP2, SAMSN1, FABP5, MIOX, NR1I3, ALDH4A1, CYP24A1, ACSM5, HNF4G, DAO, TMEM140, MGARP, CYP2AC2, DNAJC22, CYP2C23a, GPD1, GOLT1A, BCL2L15, PTPN20, SLC23A3, TMEM82, HIST1H46L2, C1orf210, AKR1E2, ABCB1, SLC13A3* | *Ichikawa et al. (2022)* |

understanding the mechanism of sex differentiation and identifying key genes involved, we can potentially improve the welfare of male chicks significantly and enhance their production efficiency. In this article, we provide a comprehensive summary of the genes explored in relation to sex differentiation of chicken embryos over recent decades. This summary serves not only as a reliable reference for researchers but also highlights potential genetic targets for further exploration. These targets could advance the development of sex screening methods in chicken embryos. While this research may not directly contribute to the advancement of these technologies, it provides foundational insights that could inspire and potentially catalyze future technological innovations in poultry production, especially in overcoming the challenge of egg sex screening.

## Further perspective

Although our current understanding is incomplete and the practical industrial applications of these findings remain limited, these insights open up transformative possibilities for future biotechnological applications.

By pinpointing specific genes, researchers can employ gene editing, mutations, or overexpression to precisely control gene expression. If proteins translated by these genes are instrumental in sex differentiation, designing antagonists or agonists could offer another layer of control. Such genetic and pharmacological strategies have significant potential in agriculture, particularly in optimizing poultry production by controlling the sex ratio. This not only improves efficiency but also has implications for conservation and livestock industries by applying similar strategies across species.

In summary, leveraging our understanding of sex differentiation genes through biotechnology could revolutionize agricultural practices and beyond, offering a streamlined approach to enhancing productivity and conservation efforts. Of course, realizing these advancements is contingent upon future researchers' comprehensive exploration of the complex mechanisms underlying chicken sex differentiation. Unraveling these intricacies holds significant promise for improving the welfare of male chicks and the applicability of production innovations.

### Funding

This work is supported by the National Natural Science Foundation of China (No.32002156), the Natural Science Foundation of Guangdong (No.2019A1515110454), the Qingyuan City Qingcheng District Science and Technology Planning Project (2020A01, 2021SJXM011), and the Design and breeding of new broiler breeds of high-quality processing types (2023ZD04064). The funders had no role in study design, data collection and analysis, decision to publish, or preparation of the manuscript.

### Grant Disclosures

The following grant information was disclosed by the authors:
The National Natural Science Foundation of China: No. 32002156.
the Natural Science Foundation of Guangdong:  No. 2019A1515110454.

The Qingyuan City Qingcheng District Science and Technology Planning Project: 2020A01, 2021SJXM011.

Design and breeding of new broiler breeds of high-quality processing types: 2023ZD04064.

## Competing Interests

The authors declare there are no competing interests.

## Author Contributions

- Xiaolu Luo conceived and designed the experiments, performed the experiments, analyzed the data, prepared figures and/or tables, and approved the final draft.
- Jiancheng Guo performed the experiments, analyzed the data, prepared figures and/or tables, and approved the final draft.
- Jiahang Zhang performed the experiments, analyzed the data, prepared figures and/or tables, and approved the final draft.
- Zheng Ma conceived and designed the experiments, prepared figures and/or tables, authored or reviewed drafts of the article, and approved the final draft.
- Hua Li conceived and designed the experiments, authored or reviewed drafts of the article, and approved the final draft.

## Data Availability

This is a literature review.

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
