# Peer review of "Overview of chicken embryo genes related to sex differentiation"

_PeerJ, doi:10.7717/peerj.17072_

## Round 0.1 · original submission · Minor Revisions

Dear Dr Kuo,
I ask you to carefully consider the comments of the reviewers, especially reviewer 3, who advised me to reject the paper and send me a corrected version of the manuscript.

Best wishes,
Alexander

**Language Note:** PeerJ staff have identified that the English language needs to be improved. When you prepare your next revision, please either (i) have a colleague who is proficient in English and familiar with the subject matter review your manuscript, or (ii) contact a professional editing service to review your manuscript. PeerJ can provide language editing services - you can contact us at copyediting@peerj.com for pricing (be sure to provide your manuscript number and title). – PeerJ Staff

Reviewer 1 ·

Basic reporting

Comments and Suggestions for Authors:
This is indeed an interesting topic, as sex determination in chicken embryos remains a central question in poultry science. This manuscript delves into the genes previously documented in connection with sex determination in chicken embryos. The examination of these genes has the potential to guide the application of gene editing technology to address the challenge of sex determination. While the article is logically structured and well organized, there are some issues that need attention from the authors.
Major reversion
1, Abstract lines 20-22, “provides a credible reference for the study of sex screening technologies on hatching eggs and for the study of gene-edited chicken embryo technology” and conclusion lines 258-259 “also provide a direction for the development of gene editing technology in chicken embryos to solve the problem of egg sex screening in production.” Understanding the genetic basis of sex determination can only identify potential gene targets for gene editing or sex screening; it cannot, however, directly enhance the technology of sex screening and gene editing.
2, The author should distinguish the sex-linked genes from others, since the sex-linked gene are most likely the sex determining genes, and others may serve as regulators of the sex determining gene.
3, This manuscript discussed numerous candidate genes, with some, including the Z-linked DMRT1, proven to be essential for sex determination in chickens. Despite these findings, the identified target genes fall short of providing a comprehensive understanding of the intricate mechanism underlying sex differentiation in chickens. Their limitations pose challenges for practical industrial applications. What are the potential avenues for advancing the application of these genes in an industrial context? To address this, a "Future Perspectives" section is necessary to delve into the authors' considerations and envisage the trajectory for applying these genes in the future.

Minor reversion:
1, Most of references are posed in intermediate sentences, so, if possible, locate at end of sentence.
2, The keywords are required.
3, Lines 86-87, a brief introduction should be included before examination of individual genes.
4, Line 115, "misexpressed" needs to be confirmed.

Experimental design

no comment

Validity of the findings

no comment

Additional comments

no comment

·

Basic reporting

The Luo et. al. summarizes the genes related to the sex diûerentiation of chicken early embryos, which provides a credible reference for the study of sex screening technologies on hatching eggs and the study of gene-edited chicken embryo technology.

Experimental design

The Review manuscript is organized logically into coherent paragraphs.

Validity of the findings

The review offers hope for enhancing animal welfare and promoting conservation efforts in poultry production.

Reviewer 3 ·

Basic reporting

This review comprehensively covered articles published in the avian sex determination and differentiation. However, most of the genes described in this review are well documented in elsewhere and novelty is less.

Although the authors insist that the present review aims to share the evidence to reduce male chicks in the poultry industry for egg production, the context of this review does not fit the objective they described.

Experimental design

This review comprehensively covered articles published in the avian sex determination and differentiation.

Validity of the findings

Most avian sex determination and differentiation-related genes in the present review are well-reviewed elsewhere.

Reviewer 4 ·

Basic reporting

The review by Luo and colleagues provides a solid and comprehensive overview of the current research on genes related to chicken embryo sex differentiation. It's well-organized, clearly written, and hits all the essential points, but a few minor things could be improved.
The authors should watch out for occasional verb tense inconsistencies and ensure everything flows smoothly in the past or present tense.

Experimental design

The review clearly outlines the methods used in the studies covered, including the search strategy, selection criteria, and analysis methods. Luo and colleagues maintain a neutral and objective tone throughout the review, avoiding personal opinions or biases.
The review focuses on genes related to sex differentiation in chicken embryos, staying within the journal's scope and avoiding irrelevant information.

Validity of the findings

The review highlights the potential impact of this research on improving animal welfare and promoting sustainable poultry production. The author should discuss the novelty of the findings and their potential to advance the field of chicken embryo sex differentiation research.

The conclusion effectively summarizes the critical points of the review and emphasizes the significance of studying genes related to chicken embryo sex differentiation. However, it could be further strengthened by providing a more forward-looking perspective and outlining the potential implications of this research for the future of the poultry industry.

Additional comments

The reference list is comprehensive and includes relevant and recent publications. However, it's always a good practice to double-check the formatting to ensure it complies with the journal's style guide.
For example, the DOI link is not available for all the references.

---

## Round 0.2 · accepted · Accept

Dear Dr. Luo,

I have assessed the revision myself, and I am happy with the current version. Thank you for the detailed responses to the comments of the reviewers and for the corrections made to the manuscript.
Now your manuscript is ready for publication.

Best regards,
Alexander